

# Heteroassociative storage of hippocampal pattern sequences in the CA3 subregion

Raphael Y. de Camargo,  Renan S. Recio and  Marcelo B. Reyes

Center for Mathematics, Computing and Cognition, Universidade Federal do ABC, São Bernardo do Campo, São Paulo, Brazil

## ABSTRACT

**Background**. Recent research suggests that the CA3 subregion of the hippocampus has properties of both autoassociative network, due to its ability to complete partial cues, tolerate noise, and store associations between memories, and heteroassociative one, due to its ability to store and retrieve sequences of patterns. Although there are several computational models of the CA3 as an autoassociative network, more detailed evaluations of its heteroassociative properties are missing.

**Methods**. We developed a model of the CA3 subregion containing 10,000 integrate-and-fire neurons with both recurrent excitatory and inhibitory connections, and which exhibits coupled oscillations in the gamma and theta ranges. We stored thousands of pattern sequences using a heteroassociative learning rule with competitive synaptic scaling.

**Results**. We showed that a purely heteroassociative network model can (i) retrieve pattern sequences from partial cues with external noise and incomplete connectivity, (ii) achieve homeostasis regarding the number of connections per neuron when many patterns are stored when using synaptic scaling, (iii) continuously update the set of retrievable patterns, guaranteeing that the last stored patterns can be retrieved and older ones can be forgotten.

**Discussion**. Heteroassociative networks with synaptic scaling rules seem sufficient to achieve many desirable features regarding connectivity homeostasis, pattern sequence retrieval, noise tolerance and updating of the set of retrievable patterns.

Corresponding author
Raphael Y. de Camargo,
raphael.camargo@ufabc.edu.br

## INTRODUCTION

Even though it is well established that hippocampal formation is responsible for temporary storage and retrieval of memories (*Squire, 1992*; *Andersen et al., 2007*), memory coding is still not completely understood. The CA3 subregion contains a particularly large number of recurrent connections among the pyramidal neurons. Existing attractor neural network models (*Hopfield, 1982*; *Amit, 1989*) show that networks with recurrent connections can use a Hebbian learning rule to store patterns, represented as a set of active neurons, and retrieve them from partial cues. Based on these models, *Rolls et al. (1997)* proposed that the CA3 subregion could work as an autoassociative memory, where neurons from the same pattern have recurrent excitatory connections between them. These connections enable associations between features of a memory, retrieval of stored memories from presentation

of partial cues, and noise tolerance during memory retrieval. These characteristics of autoassociative networks make them excellent candidates for the storage of complex episodic memories (*Rolls, 2010*), which contain smaller components organized in a flexible way and that could be used as cues to retrieve a whole episode (*Cohen & Eichenbaum, 1993*).

Another important characteristic of the CA3 subregion is that, during exploratory behavior in rats, this area shows a local field potential (LFP) signal composed of gamma (40–100 Hz) components nested in a theta (5–10 Hz) rhythm (*Bragin et al., 1995*; *Colgin, 2016*). Moreover, *O'Keefe & Recce (1993)* observed that the phase of the theta cycle at which a place cell is activated depends on the distance of the rat to the cell preferred location, an effect called theta phase precession. *Jensen & Lisman (1996)* proposed that the hippocampus could work as a heteroassociative memory (*Sompolinsky & Kanter, 1986*), a kind of network that stores sequences of patterns, represented by neuronal ensembles. Similarly, *Levy (1996)* proposed a simple model composed of McCulloch-Pitts threshold neurons that could learn to store sequences using local associative learning rules. The main difference to autoassociative memory is that neurons from a pattern have recurrent connections to neurons from the next pattern in the sequence. The advantage is that an entire sequence of patterns can then be retrieved in a theta cycle, with one pattern per gamma cycle nested in the theta cycle. However, the lack of autoassociative connections could make these network less reliable in the presence of internal and external noise and unable to perform pattern completion. To combine the properties of auto- and heteroassociative networks, models containing both kinds of connections were proposed. In these models, recurrent connections within a single CA3 area would work as autoassociative connections, while heteroassociave connections would be located either as feedback connections from CA3 to dentate gyrus (*Lisman, Talamini & Raffone, 2005*) or between different CA3 areas (*Samura, Hattori & Ishizaki, 2008*).

More recently, the concept of theta sequences (*Foster & Wilson, 2007*) was proposed, based on experimental evidence that sequences of neuronal ensembles are sequentially activated at subsequent gamma cycles and coupled to theta oscillations. Activations occur preferentially at some phases of theta cycles and sequences seem to be bounded within theta cycles. In contrast to phase precession, the appearance of theta sequences appears to require learning (*Feng, Silva & Foster, 2015*) and was associated with the representation of current goals (*Wikenheiser & Redish, 2015*) of rodents. The relationship of theta sequences with episodic (*Wang et al., 2014*) and spatial memories (*Dragoi & Buzsáki, 2006*) indicates that the CA3 recurrent and CA3-CA1 connections may encode these memories as sequential activation of neuronal ensembles.

Another feature of the hippocampus is that memories appear to be temporarily stored there and later coded as remote memories in the cortex (*Frankland & Bontempi, 2005*). This property could be due to the limited capacity of auto- and heteroassociative networks, since increasing the number of stored memories beyond its capacity leads to a catastrophic interference effect (*Amit, 1989*), where the saturation of connection weights makes all stored memories unavailable. Although the replacement of older memories by newer ones also receives the name of catastrophic forgetting in other types of networks (*French, 1999*), this effect is consistent with one of the hypothesized roles of the CA3 as a temporary

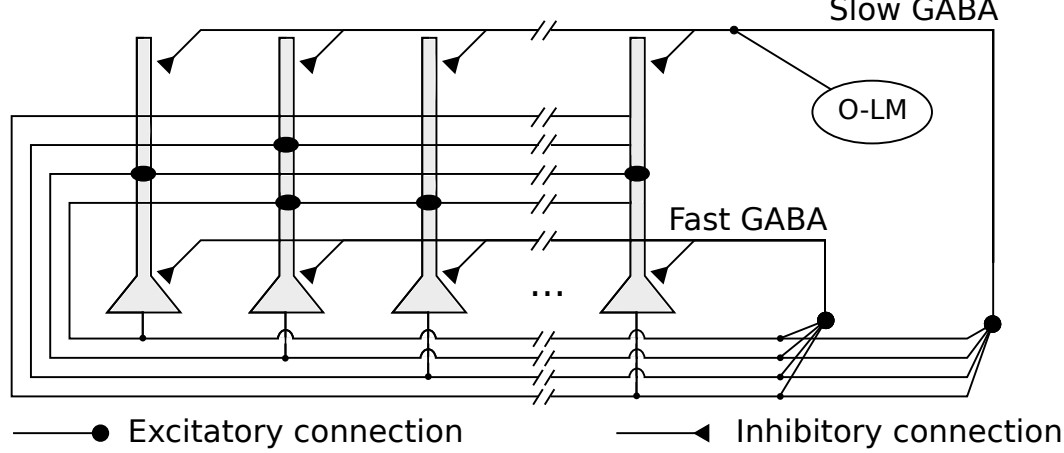

**Figure 1** Network architecture, with the recurrent connections among pyramidal neurons, fast and slow GABAergic connections, and O-LM inhibitory population.

memory storage. Proposals for enabling this replacement of older memories include the use of STDP rules (*Caporale & Dan, 2008*) and memory recall and consolidation during sleep (*Born, Rasch & Gais, 2006*).

Although there are several computational models of the CA3 as an autoassociative network (*Rolls et al., 1997*; *Cutsuridis & Wennekers, 2009*), more detailed evaluations of its heteroassociative properties are missing. In this work, we use a heteroassociative network model of the CA3 subregion containing 10,000 neurons to investigate the following properties: (i) retrieval of pattern sequences from partial cues with external noise and incomplete connectivity, (ii) achievement of homeostasis regarding the number of connections per neuron when many patterns are stored using synaptic scaling, (iii) replacement of the set of stored patterns, guaranteeing that the last stored patterns can be retrieved and older ones can be forgotten.

## METHODS

### The network model

We implemented a heteroassociative network, that simulates the CA3 subregion, composed of 10,000 integrate-and-fire neurons representing pyramidal cells. Neurons are connected by excitatory recurrent connections, with kinetic models of AMPA synaptic channels and weights defined by a heteroassociative learning rule. The network is shown in Fig. 1.

The network has two kinds of feedback inhibition, mediated by fast and slow GABA channels (*Pearce, 1993*), modeled as direct connections between every pair of pyramidal neurons. The fast inhibition is responsible for controlling the level of activity in the network and reducing the interference between stored pattern sequences. The slower inhibition helps to control the level of activity in the network in a slower time scale. We also simulated periodic inhibition from a oriens-lacunosum-moleculare (O-LM) interneuron (*Freund & Buzsáki, 1998*) population, modeling it as a single spike generator representing the intrinsic

firing rate in the theta range (5 Hz) of these cells when depolarized (*Gloveli et al., 2005*). It is connected with every pyramidal neuron via slow GABA synapses and generates the theta rhythm in the network.

Following the model used by *De Almeida, Idiart & Lisman (2007)*, we model the membrane potential $V_n$ of pyramidal neuron $n$ by:

$$\frac{dV_n}{dt} = \frac{1}{\tau_m}((I_{syn} + I_{rep}) \times r_{\text{Input}} - V_n + V_{\text{rest}})$$

using the values $r_{\text{Input}} = 33\ \text{M}\Omega$, $V_{\text{rest}} = -60\ \text{mV}$, $\tau_m = 2\ \text{ms}$. When $V_n$ reaches a threshold of $-50\ \text{mV}$, it is reset to $V_{\text{rest}}$, with a refractory period of 13.3 ms. The current $I_{rep}$ is a hyperpolarizing current that provides spike frequency adaptation and depends on the time of the last generated spike $t_{spk}$. It is given by:

$$I_{rep} = -560\ \text{pA} \times \exp\left(-\frac{t - t_{spk}}{5\ \text{ms}}\right).$$

Finally, $I_{syn}$ represents the synaptic current into the neuron, and is given by:

$$I_{syn} = I_{\text{ext}} + I_{\text{ampa}} + I_{\text{gaba}} + I_{\text{gabaS}}$$

where $I_{\text{ext}}$ represents the external input, $I_{\text{ampa}}$ the excitatory AMPA synapses, $I_{\text{gaba}}$ the inhibitory fast GABA synapses, and $I_{\text{gabaS}}$ the slow GABA synapses. We used the Euler method with an integration step of 0.1 ms, which is sufficient for simple integrate-and-fire models.

## The synaptic model

We modeled the synaptic channels using a dual exponential model, given by:

$$I_k = A_k \sum_s W_s \frac{[\Delta t_s]_+}{\tau_1 - \tau_2}\left(\exp\left(-\frac{\Delta t_s}{\tau_1}\right) - \exp\left(-\frac{\Delta t_s}{\tau_2}\right)\right)$$

where $[\Delta t_s]_+ = t - t_s - t_{\text{delay}}, \forall t > t_s + t_{\text{delay}}$ is the time since the spike $s$ generated at time $t_s$ was delivered considering the delay $t_{\text{delay}}$, and 0 otherwise. $A_k$ represents the maximum conductance and $W_s$ the synaptic weight, i.e., the connection strength between the neurons. Parameters $\tau_1$ and $\tau_2$ are the time constants, and the double exponential is reduced to the alpha function when $\tau_1 = \tau_2$:

$$I_k = A_k \sum_s W_s \frac{[\Delta t_s]_+}{\tau_1} \exp\left(1 - \frac{\Delta t_s}{\tau_1}\right).$$

We used the $\tau_1 = 2\ \text{ms}$, $\tau_2 = 8\ \text{ms}$ and $A_k = 3{,}200\ \text{pA}$ for AMPA channel of recurrent connections (*Spruston, Jonas & Sakmann, 1995*), $\tau_1 = \tau_2 = 2\ \text{ms}$ and $A_k = 3{,}200\ \text{pA}$ for the external input channels, $\tau_1 = \tau_2 = 5\ \text{ms}$ and $A_k = 540\ \text{pA}$ for the fast GABA channels (*Pearce, 1993*), and $\tau_1 = 7\ \text{ms}$, $\tau_2 = 57\ \text{ms}$ and $A_k = 30\ \text{pA}$ for the slow GABA channels (*Pearce, 1993*). We did not include NMDA channels, which have a main role in LTP and LTD processes, since our learning phase is executed offline, as we discuss in subsection "Learning rule".

We placed all neurons in a plane and defined the axonal delay between each pair of pyramidal neurons based on their euclidean distance. We used an axonal propagation

velocity of 300 μm/s (*Meeks & Mennerick, 2007*) and considered a square 2 × 2 mm area, based on anatomical data (*Amaral & Witter, 1989*), resulting in a mean axonal delay of about 3.3 ms (*Aaron & Dichter, 2001*). Since we are using integrate-and-fire neurons, we also included the excitatory postsynaptic potential (EPSP) propagation time to the soma. We used a delay of 5 ms, estimated by experimentally measured differences between the time to peak of EPSPs in the soma from mossy fiber inputs, near the soma, and recurrent connections, at apical dendrites (*Miles & Wong, 1986*).

For inhibitory feedback connections we used a delay of 2.5 ms, based on studies on interneurons from the CA3 (*Diego et al., 2001*). For the slow inhibition, we used a delay of 10 ms, to include the fact that these inhibitory cells are located near the basal positions of the pyramidal cells and makes connections to the upper apical positions (*Hájos et al., 2004*).

External noise was modeled as a Poisson process with 1 Hz rate per neuron, representing independent inputs to the neurons not related to the retrieval of sequences. We used weights that were large enough to cause the network neurons to fire spontaneously at the rate of 0.75 Hz, which is similar to the rate 0.80 Hz produced during pattern retrieval in the simulations.

## Pattern sequences

We created the patterns by generating random permutations, without replacement, of the neuron indexes and getting the first $r$ values of the permutation, where $r$ is equal to the number of neurons times the pattern density for the simulation. We define the density as the number of neurons active in a pattern divided by the total number of neurons. For heteroassociative networks, there are estimates by *De Almeida, Idiart & Lisman (2007)* of density of 0.003. For our simulated network with 10,000 neurons this would result in only 30 neurons per pattern, which would be insufficient for noise tolerance and sequence completion. We use the term sequence completion to refer to that the next pattern in the sequence can be retrieved correctly even when the current pattern was only partially activated. Although using more neurons per pattern decreases the storage capacity of the network, we decided to use a density of 0.01, resulting in 100 neurons per pattern.

We define a pattern $p^\mu$ as a binary vector with size equal to number of pyramidal neurons in the network. Each position $p_i^\mu$ has value 1 if neuron $i$ is part of $p^\mu$ and 0 otherwise. We considered sequences of seven patterns, which appears to be the typical number of patterns that can be coded inside theta cycles (*Lisman & Idiart, 1995*; *Lisman & Jensen, 2013*), and stored them using a heteroassociative learning rule (*Sompolinsky & Kanter, 1986*) that associates each pattern in the sequence with the subsequent one. For example, in a sequence of patterns s1–s2–s3–s4–s5–s6–s7, neurons from pattern s1 will have excitatory connections to neurons from pattern s2, which in turn connect to neurons from s3, and so on. We included additional connections from the last to the first pattern, permitting sequence retrieval using a cue from any of its patterns.

## Learning rule

The model uses a heteroassociative learning rule (*Sompolinsky & Kanter, 1986*) for pattern sequence storage. We simulated an offline learning procedure, performed before the

simulation, by considering that the patterns from each sequence were presented in sequence. This enables the usage of an LTP-like rule to learn associations between subsequent patterns in each sequence, which can be summarized as:

$$\Delta w_{ij} = \gamma^+ p_i^\mu p_j^{\mathrm{mod}(\mu+1,k)}$$

where $w_{ij}$ represents the weight of the connection from neuron $i$ to neuron $j$, $\gamma^+ = 1$ the learning rate, $p_i^\mu$ the neuron $i$ from pattern $\mu$ and $p_j^{\mathrm{mod}(\mu+1,k)}$ the neuron $j$ from the next pattern in a sequence of $k$ patterns, $\mu = 0, \ldots, k-1$. The function $\mathrm{mod}(x,k)$ represents the modulo operation, with $\mathrm{mod}(-1,k)$ defined as $k-1$. This produces an association between the last and the first patterns in the sequence. We also used an LTD-like learning rule that performs a dissociation between a pattern and the previous pattern in the sequence, summarized as:

$$\Delta w_{ij} = -\gamma^- p_i^\mu p_j^{\mathrm{mod}(\mu-1,k)}.$$

We executed simulations with and without LTD, in which case we used $\gamma^- = 1$ and $\gamma^- = 0$, respectively.

We included synaptic scaling (*Abbott & Nelson, 2000*) as a heterosynaptic long-term depression (hLTD) mechanism (*Chistiakova & Volgushev, 2009*), where we update the weights of synapses that were not active during the learning phase. Synaptic scaling models a competition between the synapses of a single neuron. After learning a set of 100 pattern sequences, we apply an additive synaptic scaling rule to all neurons. We decrease all synaptic weights from each neuron by a fixed amount, defined as the ratio between the total weight increase since the last scaling procedure and the number of connections. Synapses that become negative are set to zero and the difference is uniformly subtracted from the remaining synapses.

In biological networks, anatomical constraints can limit the set of neurons to which a neuron can connect. We simulate these constraints using different levels of initial connectivities, where we randomly defined for each neuron the set of neurons to which it can connect. For an initial connectivity of 0.6, each neuron initially connects to 60% randomly selected neurons in the network, with random weights drawn from a uniform distribution between 0 and a defined maximum value. Initial connection weights are scaled to maintain the excitatory drive with different initial connectivities comparable. Connections to neurons outside the set of initial connections cannot be created during learning.

The learning rule is then sequentially applied to each pattern sequence to be stored. During learning connections are eliminated due to LTD or synaptic scaling, so that after storing many sequences the network stabilizes at a lower connectivity, effectively simulating a "lifetime" of episodic sequences learning until maturity.

## Sequence retrieval

We defined as a pattern cue the stimulation of a subset of the neurons that are active in the pattern. We always provide inputs for same number of neurons, so that with a cue size of 1.0, we stimulate all neurons from the pattern and no other neuron. For instance, with

a cue size of 0.7, only 70% of the neurons in the pattern receive the input, while random neurons outside the pattern are selected to keep the input size controlled over different simulation runs. We consider that during retrieval the entorhinal cortex (EC) provides the input to the CA3 via the perforant path, using a coding where an ensemble of neurons, representing an input pattern, fire synchronously inside a single gamma cycle (*Chrobak & Buzsáki, 1998*). In this case, each neuron stimulated by the cue receives a single spike as input, with a synaptic strength large enough to cause its firing. A second input to the CA3 are the mossy fibers, originating from the dentate gyrus, which appears to deliver sparse patterns for storage in the CA3 subregion (*Treves & Rolls, 1992*). Since in this work the learning phase was performed offline, we did not include this input in the simulation.

We determine the set of active neurons by checking which neurons fired inside a time window of 10 ms. The measurements are performed every 2 ms with the time window centered at simulation time $t$. We used a small time window to permit capturing active neurons only from a single gamma cycle at each time $t$ in the middle of the cycles. We define the *overlap* of the active neurons with a stored pattern $p^\mu$ as:

$$\text{overlap}(t, p^\mu) = \frac{1}{n_{p^\mu}} \sum_{i \in p^\mu} o_i(t) \, p_i^\mu$$

where $n_{p^\mu}$ is the size of $p^\mu$, $o_i$ is the state of neuron $i$ at the instant $t$, which can be 0 (inactive) or 1 (active), and $p_i^\mu$ is 1 if neuron $i$ is part of $p^\mu$ and 0 otherwise. The overlap equals 1 when all the neurons from a stored pattern are active. Neurons also have spontaneous firings due to external noise and, to correctly evaluate the pattern retrieval, we consider in the metric only the neurons which are part of the pattern. This could be applied because the number of active neurons was always comparable to the number of neurons per pattern. To verify that extra active neurons are randomly activated and not concurrent retrieval of multiple patterns, we also compute the highest overlap among all other patterns.

We evaluated sequence retrieval in the simulated neuronal network using the mean overlap over 10 simulations for each combination of parameter values. We varied the cue size (1.0, 0.8, 0.6 and 0.4), initial connectivity (1.0, 0.8, 0.6 and 0.4) and synaptic LTD usage (true and false). We stored 10,010 patterns, a multiple of the sequence size of 7 patterns, in the network. Each simulation was executed for 86,150 ms, which was sufficient to evaluate the retrieval of the last 3,003 stored patterns. The simulation and experimental results analysis were implemented and executed in MATLAB (R2014a; The Mathworks, Natick, MA, USA). Simulation source code and raw and processed data are available at a public repository (https://dx.doi.org/10.6084/m9.figshare.5378347.v2).

## RESULTS

### The number of retrievable patterns and connections are stable during the network lifetime

Our evaluation shows that the number of connections in the recurrent network, initialized with full connectivity and random weights, stabilizes after learning a few hundred patterns, remaining stable as new patterns are added (Fig. 2A). Storing a pattern sequence increases the total synaptic weight of each neuron, which is compensated by the synaptic scaling

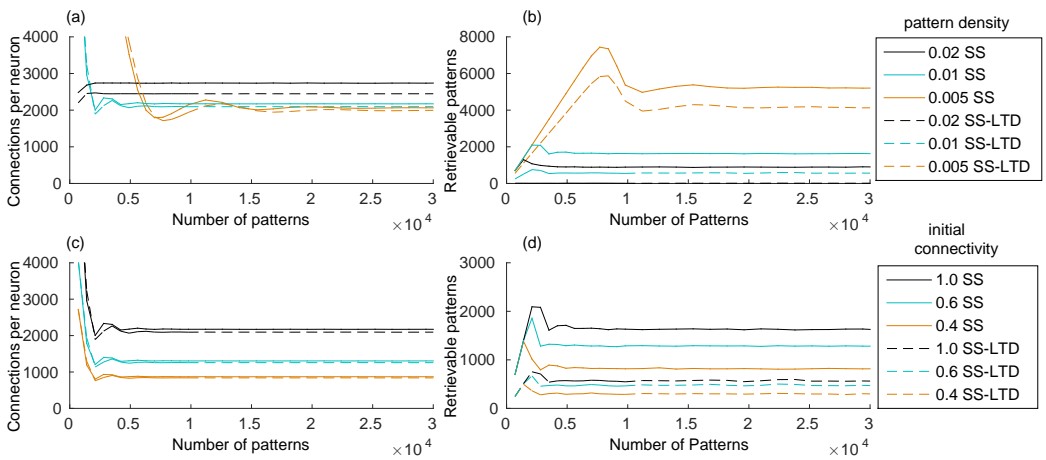

**Figure 2** (A) Number of connections in a network with 10,000 neurons after learning a set of pattern sequences when applying only the synaptic scaling (SS) rule, for a pattern density of 0.02, 0.01 and 0.005, and with synaptic scaling and LTD (SS–LTD); (B) number of retrievable patterns for the same scenarios from (A); (C–D) same as (A–B), but comparing neuronal initial connectivities of 1.0, 0.6 and 0.4. The graph shows the mean of three executions. Standard deviations values were lower than 10 connections for (A) and (C), 70 patterns for (B) and 35 patterns for (D) and are not shown for clarity.

rule. This reduces the weights of all synapses in the neuron, causing some of them to disappear as they approach zero. The number of connections decreases until the number of connections created by learning new patterns is balanced by the removal of old connections. This occurred either when using only synaptic scaling (SS), in which case we set $\gamma^- = 0$, and when using SS combined with LTD (SS–LTD), with $\gamma^- = -1$, showing that synaptic scaling is sufficient to control the number of connections in the network.

The number of retrievable patterns initially increases with the number of stored patterns, followed by a decrease toward a constant value, as shown in Fig. 2B. To determine if a pattern $p^\mu$ is retrievable, we compute the sum of excitatory weights $S_i$ from neurons from the previous pattern $p^{\mu-1}$ toward each neuron $i$ in the network. If $S_i$ for each target neuron from $p^\mu$ is larger than the $S_i$ of each neuron outside $p^\mu$, the pattern is considered retrievable. This means that neurons in the pattern receive a larger excitatory drive than the other neurons during pattern retrieval, which permits a global inhibition to be used to suppress the activity of these other neurons. This simple measurement generates an estimate of the number of retrievable patterns directly from the connection weights. The alternative would be a simulation where, for each pattern sequence, we would provide a cue of one of its patterns and check if all patterns in the sequence are retrieved correctly. But this would be unfeasible, since we would have to perform simulations to retrieve every pattern for all network configuration combinations, where we would vary the number of stored patterns, pattern density level, initial connectivity and LTD rule.

The initial increase in number of retrievable patterns (Fig. 2B) is due to the lack of patterns to retrieve in the beginning and the later decrease occurs due to the interference between stored patterns, called crosstalk. After reaching an equilibrium point, the number

of retrievable patterns remains constant. Using a pattern density of 0.02, instead of 0.01, increased the number of connections, causing an increase in the crosstalk between patterns, which resulted in a reduced number of retrievable patterns (Figs. 2A–2B). Combined with LTD, which caused a larger number of weight decreases, the number of retrievable patterns became almost zero. With density 0.005, the final number of connections was smaller and the number of retrievable patterns higher, showing the benefits of using fewer neurons per pattern. Although a smaller density may reduce the reliability of the network due to neuron firing and synaptic variability, a region containing one million neurons would still have about 5,000 active neurons per pattern with a density of 0.005.

Including LTD (SS–LTD) caused only small changes in the number of connections, but the number of retrievable patterns decreased (Figs. 2A–2B). Although this result might seem to occur due to extra reductions in excitatory weights caused by LTD, we compensate for this by reducing the effect of synaptic scaling, so that the total weights per neuron with or without LTD will be the same. It appears that using LTD causes disruptions in stored pattern sequences when decreasing the connection weights from neurons of the last stored pattern to neurons of the previous pattern.

Anatomical constraints may prevent full connectivity in the CA3 subregion. An initial connectivity of 0.6 means that a neuron can connect to 60% of the neurons. Using a smaller initial connectivity reduces the number of connections, but the number of retrievable patterns is also reduced (Figs. 2C–2D). However, it is interesting that using an initial connectivity of 0.4 and synaptic scaling (SS) alone, there were about 900 retrievable patterns, which is more than the 600 patterns obtained when using LTD (SS–LTD) with full initial connectivity. Also, although 900 patterns may be too few, this value would probably be higher for larger networks from biological systems.

Increasing the initial total weight per neuron increases the number of connections, caused by less competition in the synaptic scaling rule (Fig. 3A). It also increases the number of retrievable patterns, as old patterns take longer to be forgotten, until a transition point where the crosstalk interference becomes larger than the pattern inputs (Fig. 3B). After this point, the number of retrievable patterns starts to decrease. The behavior is similar both with and without the usage of LTD, but in the latter the number of retrievable patterns was larger.

Another way to analyze the results is to check how many patterns can be stored in a network with a given number of connections per neuron. For instance, *De Almeida, Idiart & Lisman (2007)* estimated that neurons in the CA3a subarea connect to 20% of the neurons. In our network, it would result in about 1,600 retrievable patterns and would be far from the transition point. Including more neurons or using a smaller pattern density, as is the case in biological systems, would result in a much larger number of retrievable patterns.

The final number of connections in the network is dependent on the initial random weights between neurons, initial connectivity, and pattern density (Figs. 3A and 3C). With larger initial weights, the sum of synaptic weights toward each neuron will be larger and, consequently, fewer connections will be removed by the effect of synaptic scaling and LTD.

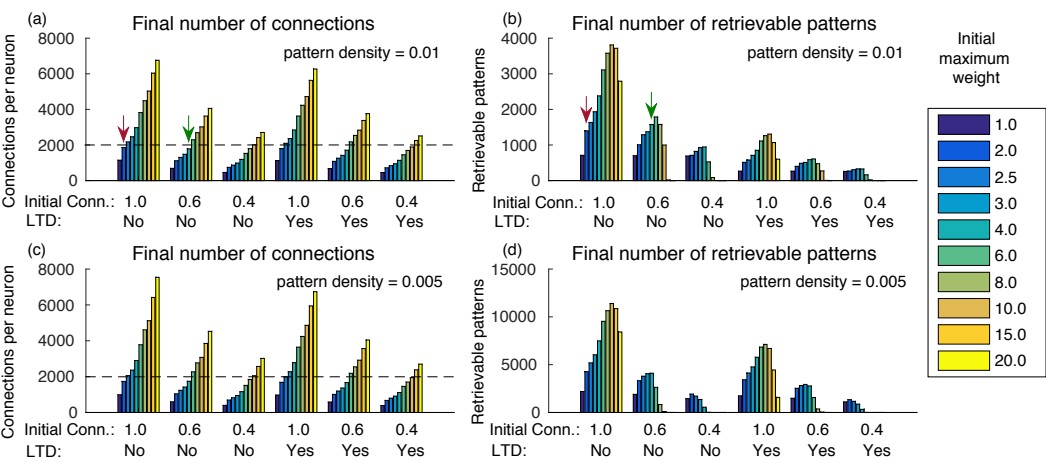

**Figure 3** **(A) Number of connections after learning 100,000 patterns and applying the synaptic scaling rule for different total weights per neuron, for a pattern density of 0.01, initial connectivities of 1.0, 0.6 and 0.4 with and without LTD; (B) number of retrievable patterns for the same scenarios from (A); (C–D) same as (A–B), but with density 0.05.** The graph shows the mean of three executions. Standard deviations values were lower than three connections for (A) and (C), 32 patterns for (B) and 63 patterns for (D) and are not shown for clarity.

It is also interesting that although a smaller initial connectivity would permit the storage of more patterns with the same number of connections, we actually see a reduction in the number of retrievable patterns (Figs. 3A and 3B). This is caused by the smaller signal-to-noise ratio in the system, as there will be proportionally fewer coding connections. If we consider an estimated final connectivity of 20% (*De Almeida, Idiart & Lisman, 2007*), with full initial connectivity we would have to use an initial weight of 2.0, resulting in about 1,400 retrievable patterns (red arrows). But with an initial connectivity of 60%, it is possible to use a larger initial weight of 6.0, while maintaining the total connections to 20% of the neurons, which resulted in about 1,600 retrievable patterns (green arrows), an actual improvement over the full initial connectivity. With an initial connectivity of 40%, the number of retrievable patterns started to decrease with more than 1,200 connections per neuron.

Using a density of 0.005 caused small changes in the final number of connections, but the number of retrievable patterns increased (Figs. 3C–3D), which would be more typical for larger biological networks with lower density. The exception is for the initial connectivity of 0.4, due to the small number connections per neuron in the simulated network.

In this section we evaluated the static properties of the connections and the number of retrievable patterns. In the following sections, we perform actual network simulations to evaluate the dynamic operation of the network. In all simulations, we used a density of 0.01, with 100 neurons per pattern, to improve the reliability of the simulations. We used different initial weight values for different initial connectivities (weight, connectivity): (2.5, 1.0), (3.0, 0.8), (6.0, 0.6) and (4.0, 0.4). These are the values that maximize the number of retrievable patterns when neurons connect to approximately 20% of all neurons.

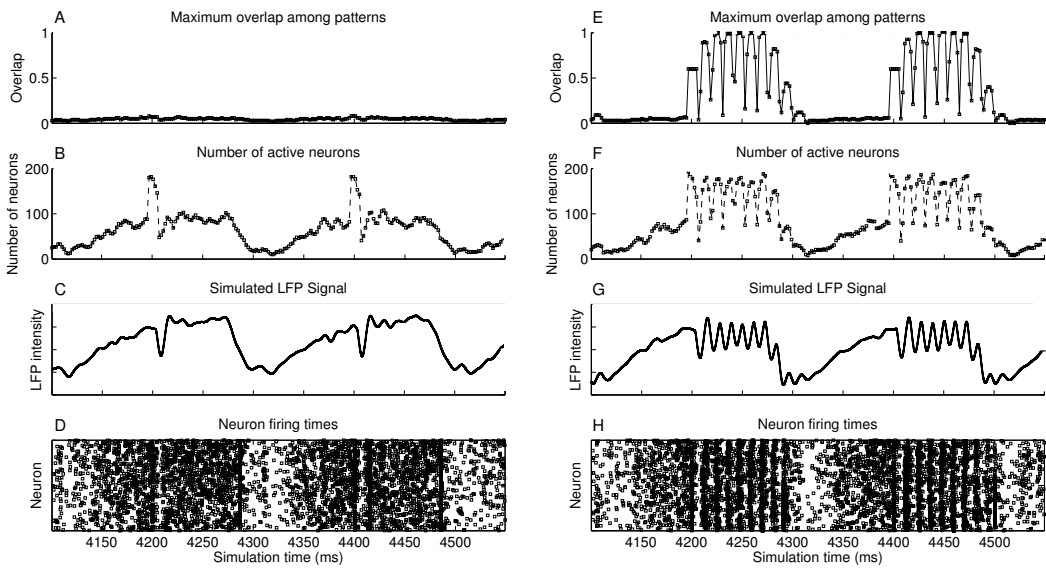

**Figure 4** Operation of two heteroassociative networks with $10^4$ pyramidal neurons and $10^4$ stored patterns with 100 neurons per pattern and coded into sets of seven patterns. The left-hand side graphs show 450 ms of network operation with presentation of random inputs at instants 4,200 ms and 4,400 ms, extracted from a 5,000 ms simulation. (A) Maximum value of the overlap among all stored patterns; (B) number of active neurons as a function of time; (C) simulated LFP signal from the network; (D) raster plot showing instants of neuronal spikes for all neurons in the network; and (E–H) same as (A–D), but with the presentation of partial cues of 0.6 of a pattern at instants 4,200 ms and 4,400 ms.

## A heteroassociative network enables the retrieval of stored pattern sequences in the presence of noise

The simulated heteroassociative network could retrieve complete pattern sequences stored on its connection weights after presentation of a partial cue from any pattern in the sequence. Although this retrieval is expected in a noiseless environment, we considered the scenario with a level of noise that caused the neurons to fire spontaneously, without presentation of pattern cues, with a mean spike rate of 0.75 Hz. The network oscillates in the theta band at 5 Hz, shown in the approximate LFP signal obtained by sum of the membrane potential of the pyramidal neurons (Fig. 4C). This oscillation in 5 Hz is due to the periodic inhibition of O-LM cells, with a sharp decrease in the spike rate when the inhibition starts and a gradual recovery of the spike rate when the inhibition fades. Also, no pattern is retrieved during this period (Fig. 4A), despite the firing of near 200 neurons at some points (Fig. 4B), showing that the firing was random, which is also illustrated in the raster plot (Fig. 4D). When a random external input is presented at instants 4,200 ms and 4,400 ms, the target neurons are activated, but since they do not represent a stored pattern, this extra activity disappears.

Presenting a pattern cue caused the retrieval of the complete pattern sequence just after its presentation, at 4,200 ms and 4,400 ms in Figs. 4E–4H. Each pattern is retrieved during a small time window, corresponding to a gamma cycle inside the theta cycle, as occurs in the theory proposed by *Lisman (2005)*. There are about nine gamma cycles per theta cycle

and since each pattern sequence has seven patterns, the complete sequence is retrieved during a single theta cycle. The last pattern in the sequence is connected to the first, which causes the number of gamma cycles to be larger than the sequence size. We can also see that although we present only 60% of the first pattern of the set (cue size of 0.6), the network successfully retrieved and completed the subsequent patterns. Finally, the raster plot of neuronal spikes shows the clustering of neuronal firings during each pattern retrieval, with periods of inactivity between gamma cycles.

Longer or smaller pattern sequences can also be stored and retrieved. With a larger sequence, a subset of the pattern sequence is retrieved in a single theta cycle, until the slow inhibitory current starts to win the competition against the excitatory drive. The remaining patterns can then be retrieved in the next theta cycle, by presenting the last pattern retrieved in the previous cycle. With smaller pattern sequences, the theta cycle would just keep looping the patterns in the sequence until the inhibition surpasses the excitatory drive and finishes the theta cycle.

In simulations shown in Figs. 4E–4H, we presented the pattern cues at the beginning of the theta cycles. Presenting the cues at other phases would cause only part of the sequence to be retrieved, since the periodical inhibition from O-LM cells would cause the pyramidal cells to stop firing before the end of the sequence. Since phase synchronization may be required for memory retrieval in the hippocampus (*Fell & Axmacher, 2011*), it seems reasonable to provide the inputs at specific theta phases.

Sequence completion during retrieval occurs due to the heteroassociative connections from pattern $j$ to next pattern $j+1$. Since many neurons from pattern $j$ must fire to activate pattern $j+1$, randomly firing neurons do not cause random pattern retrievals. Sequence completion also works with initial connectivity of 0.6, as shown in Fig. 5A. Also, the fast inhibition guarantees that only a single pattern is retrieved at a time. After evaluating the overlap with all other stored patterns, the second highest overlap was always lower than 0.1 (Fig. 5B). Finally, using LTD did not cause perceptible changes in the mean overlap values (Figs. 5C–5D).

## The heteroassociative network forgets older patterns while learning new ones

The heteroassociative network model with the synaptic scaling rule replaces old pattern sequences with newer ones. It is a required property for any model of the CA3, since just increasing the number of stored patterns would saturate synaptic weights, causing all stored pattern sequences to be lost, an effect called catastrophic interference (*Amit, 1989*). It is also in accordance with the theory of temporary storage of memories in the hippocampus, which are later transferred to the cortex as remote memories (*Frankland & Bontempi, 2005*).

The number of patterns that can be retrieved from the network depends on the cue size, as more complete cues permit the retrieval of patterns with weaker connections. Pattern sequences were stored before the simulation and the sequences were retrieved in ascending order, with one sequence retrieval per theta cycle. Figure 6A shows the retrieval of the last 3,000 stored patterns for different cue sizes. The last stored patterns are fully retrievable but, as we go backwards in the set of stored patterns, we report a strong transition point
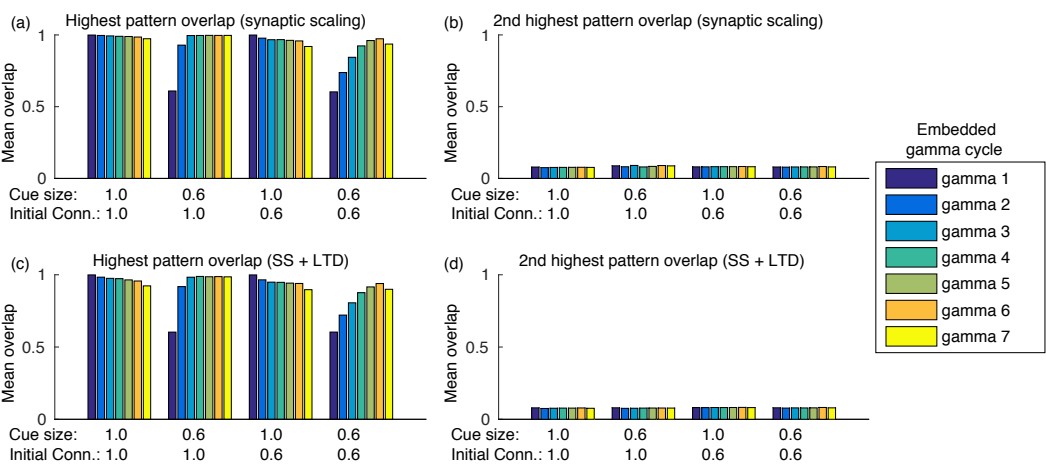

**Figure 5** **Mean overlap during pattern retrieval within each gamma cycle inside a theta cycle.** Gamma 1 refers to the first gamma cycle in each theta cycle, gamma 2 to the second, and so on. (A) The highest overlap among patterns for different cue sizes and initial connectivities and using synaptic scaling only; (B) same as (A), but showing the second highest overlap among all patterns; (C) and (D) same as (A) and (B), but using synaptic scaling and LTD. Graphs show the average mean overlap for 10 simulations. Standard deviations values were lower than 0.015 and are not shown for clarity.

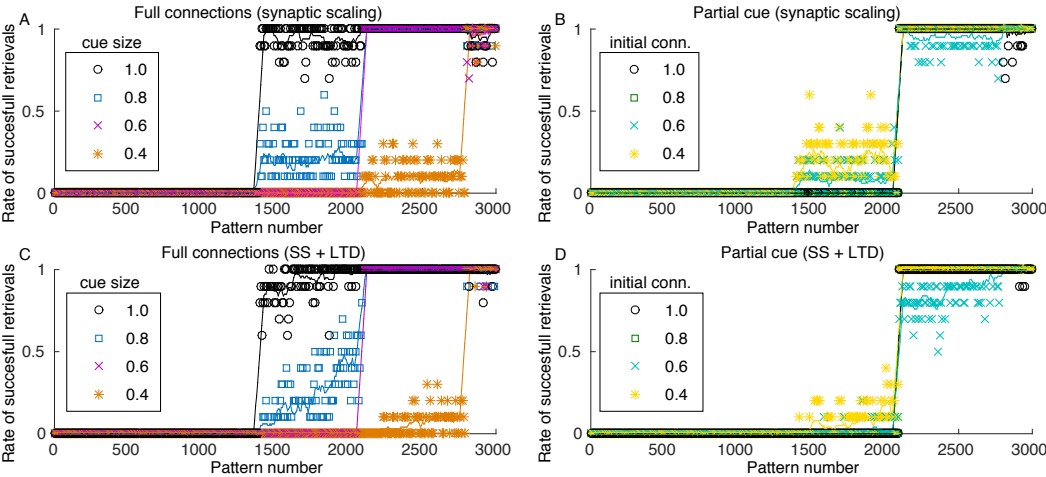

**Figure 6** **Rate of successful retrieval for the last 3,000 stored patterns.** Markers show the average retrieval rate for 10 simulations, lines represent the moving average of 10 retrieval rates and standard deviations are not shown for clarity. A retrieval is successful when at least four patterns of the sequence were retrieved with an overlap larger than 0.5. (A) Simulation with all connections, variable cue sizes, and synaptic scaling; (B) simulation with partial cues of 0.6, varying initial connectivity and synaptic scaling; (C) and (D) same as (A) and (B), but with the addition of LTD.

where patterns are no longer retrievable. The transition point depends on the cue size, with larger cues enabling the retrieval of more patterns.

The minimum cue size required to successfully retrieve the sequences was 0.4, which is related to the configuration of excitatory weights between patterns. Although with larger

weights retrieval would be possible with a smaller cue, these larger weights would also make the network more unstable. Since performing fine tuning of the network parameters and evaluating the impact of each are outside the scope of this work, we used a value that offered a good trade-off between stability and cue size. Nonetheless, the network stored between 200 and 1,600 patterns, depending on the cue size. Although this value seems small, we used a network of only 10,000 neurons, when compared to 300,000 from the rat CA3 (*Rolls, 2007*) or 2.7 million from the human CA3 (*Simic et al., 1997*).

Pattern retrieval depends weakly on the initial connectivity (Fig. 6B). Although with smaller connectivities there would be fewer connections from a retrieved pattern $j$ to neurons from pattern $j + 1$, these connections are stronger. The transition point between retrievable and forgotten patterns is actually the same, even for an initial connectivity of 40%. Consequently, even if most neuronal pairs from patterns $j$ and $j + 1$ cannot create connections due to anatomical or physiological restrictions, the network can still operate. This also permits the network to use fewer connections for pattern sequence storage, saving scarce metabolic resources in the network.

When including LTD, the capacity of the network is slightly reduced (Figs. 6C and 6D), but the effect is not as severe as indicated in Fig. 2. Also, although the capacity is reduced, the stored sequences are still retrieved correctly, as shown in Figs. 5C and 5D. These results indicate that, for the proposed heteroassociative network, an additive synaptic scaling is sufficient for replacing old memories by new ones, as the addition of LTD does not cause perceptible changes in the retrieval capability of the network.

## DISCUSSION

### Coding of memories as pattern sequences

Episodic memories are characterized by their compositional properties (*Cohen & Eichenbaum, 1993*; *Henke, 2010*) and coding as pattern sequences allows different stimuli to be combined in a single composition. The relationship of theta sequences with spatial (*Dragoi & Buzsáki, 2006*) and episodic memories (*Wang et al., 2014*), the requirement of learning for their appearance (*Feng, Silva & Foster, 2015*) and their association with the representation of current goals (*Wikenheiser & Redish, 2015*) provide important evidence for heteroassociative coding in the hippocampus.

Retrieval of pattern sequences from small cues is possible with both auto- and heteroassociative networks. This retrieval occurs, for example, when mice need to retrieve stored spatial memories from the presentation of a small cue of the original memory (*Nakazawa et al., 2002*). With heteroassociative coding we can present a small cue from a single pattern in the sequence, and the rest of the sequence would be retrieved sequentially. With autoassociative networks, neurons from the stored pattern would be progressively activated by the self-reinforcing dynamics of the network until the complete pattern is retrieved. Also, both types of network permit the storage of memories of different sizes. With heteroassociative networks, larger complex memories could be coded in longer sequences, while simpler ones could use small sequences.

The hippocampus is considered to mediate some cognitive functions related to memory, such as associative representation of events, sequential organization, and relational

networking (*Eichenbaum, 2004*). Interestingly, heteroassociative coding may address these issues. Associative representation of events can be accomplished by including events in the same pattern sequence, while a sequential organization arises naturally from pattern sequences. Relational networking considers that the same event may be part of multiple memories, which could be accomplished putting the same event in multiple pattern sequences. In this case, an extra mechanism would be required to disambiguate between these sequences during pattern sequence retrieval. For instance, *Levy (1996)* proposed that a set of context neurons in the CA3 could guide this disambiguation.

One possible problem with heteroassociative coding is that patterns cannot be held as persistent activity for an extended period of time. This activity persistence is normally considered a requirement for memory retrieval and can be accomplished with the usage of autoassociative connections. This persistent activity is normally found in the prefrontal cortex (*Fuster & Alexander, 1971*; *Funahashi, Bruce & Goldman-Rakic, 1989*), but also in other areas, such as the entorhinal cortex (*Egorov et al., 2002*). These regions are largely connected to the hippocampus, so we can envision that the hippocampus is responsible for the associations, sequential organization, and relationships (*Eichenbaum, 2004*) between events. The retrieval of individual patterns in the sequence then causes the persistent activity associated to memories in the different cortex regions.

## External noise, sequence completion and limited connectivity

The usage of heteroassociative connections in the recurrent CA3 connections is not a new subject. For instance, *Yamaguchi (2003)* studied a model that tries to explains theta phase precession using heteroassociative connections from CA3 to CA1 and in the CA3 recurrent connections. Other studies also used simple feedforward network models that associate pairs of patterns (*Lytton, 1998*; *Miyata, Ota & Aonishi, 2013*). But these studies did not evaluate the problem of sequence completion, noise tolerance and operation with limited connectivity.

Other existing heteroassociative models of the hippocampus use autoassociative connections in local circuits of the CA3, to reduce the effects of noise and incomplete pattern retrieval. But the same set of CA3 recurrent connections cannot store both autoassociative and heteroassociative connections for a given pattern, since this would provide two possible routes for the network dynamics (*Rolls & Kesner, 2006*). A solution proposed in some theoretical studies is that the autoassociative and heteroassociative connections are present in different parts of the the CA3 (*Samura, Hattori & Ishizaki, 2008*) or in connections from the CA3 to other areas of the hippocampus (*Lisman, Talamini & Raffone, 2005*). In both theoretical approaches, the heteroassociative connections perform the storage and retrieval of sequences, leaving to the autoassociative connections the task of removing noise and performing pattern completion during sequence retrieval.

We showed that a heteroassociative network can operate correctly in the presence of external noise and limited synaptic connectivity, since retrieval errors in single patterns do not propagate to the next patterns. These errors include partial pattern retrieval and activation of neurons which are not part of the retrieved pattern. Consequently, the presence of both autoassociative and heteroassociative connections in the same network may not be

a requirement for preventing these propagating errors. One limitation of the current study is that we did not include noise in the neural responses or in synaptic plasticity. But since we showed that the network operates correctly with limited connectivity, it seems that even though this extra noise would degrade the network performance, it would not prevent the network from operating properly. Nonetheless, if the objective is to obtain extra reliability, a combination of autoassociative and heteroassociative connections may be helpful.

Synfire chains (*Abeles, 1991*) are defined as the sequential activation of pools or layers of neurons by feedforward connections. It is a broader concept that encompasses many types of network architectures. Although feedback connections may be considered in randomly connected networks (*Abeles, Hayon & Lehmann, 2004*), the focus is on feedforward networks (*Tetzlaff, Geisel & Diesmann, 2002*). Nevertheless, one may consider the retrieval of pattern sequences as a synfire chain.

Recent work has shown that 8.2% of the measured connections (12 out of 146) between pyramidal neurons in the CA3 of a rodent were part of reciprocal connection motifs (*Guzman et al., 2016*). Although reciprocal connections are normally considered as a signature of autoassociative networks, they are also present in heteroassociative networks. For instance, in our simulations of heteroassociative networks, more than 13% of connections were part of reciprocal motifs. Although the percentages of reciprocal motifs found in our simulations and in the rodent CA3 cannot be directly compared, the low number of motifs in the rodent CA3 is an evidence that heteroassociative coding may be used in the CA3 area.

## System homeostasis and catastrophic forgetting

It is well known that an increase in the number of stored memories in an auto- or heteroassociative network leads to a catastrophic interference effect (*Amit, 1989*), where the saturation of connection weights makes all stored memories unavailable. It is possible to prevent catastrophic forgetting in Hopfield networks (*Hopfield, 1982*). It requires changing the learning rule to generate strong enough synaptic changes to ensure the retrieval of newly stored patterns and bounding the maximum values of synaptic weights (*Nadal et al., 1986*). This type of memory was called palimpsest memory and permits the retrieval of the last stored memories, but it can store only a small number of patterns. We note that the replacement of old memories with new ones is also reffered to as catastrophic forgetting in other types of networks (*French, 1999*). It is an important concern in procedural learning, since old learned skills may be useful and should not be indiscriminately erased by learning new ones. For the CA3 this effect is not a real problem due to the temporary nature of memory storage.

Enabling the storage of new memories requires the removal of old memories that are never recalled. According to the remote memories hypothesis, memories are temporarily stored in the hippocampus to be later coded as remote memories in the cortex (*Frankland & Bontempi, 2005*). Some form of unlearning is required, and there are several proposals on how this could be performed specifically during sleep. The two main hypotheses for this unlearning process are: (i) the active system consolidation (*Diekelmann & Born, 2010*), where memories in the hippocampus are replayed during sleep and transferred over to

the neocortex causing the potentiation of synapses, and (ii) the synaptic homeostasis hypothesis (*Tononi & Cirelli, 2014*), which states that synaptic weights are strengthened by learning during wake periods and decreased during sleeping.

Our model is compatible with both hypotheses. With the synaptic homeostasis hypothesis (*Tononi & Cirelli, 2014*), our learning of new associations would represent the wake periods and the additive synaptic scaling rule would account for the homeostatic process of reducing synaptic weights. We use synaptic scaling to maintain the total sum of weights constant for each neuron, by decreasing all synaptic weights by the same amount. Memories are forgotten as their respective synapses have their weights reduced and weaker synapses are eventually eliminated as their weights vanish.

When considering the active system consolidation (*Diekelmann & Born, 2010*), stored memories would be recalled and transferred over to the neocortex during sleep. Different mechanisms could be used for maintaining the system homeostasis, including the use of STDP rules (*Caporale & Dan, 2008*) or activity-dependent synaptic scaling (*Abbott & Nelson, 2000*). In our model, we used a synaptic scaling based on the sum of the weights, since we did not simulated the learning process itself. Synaptic scaling rules are normally based on the mean activation of the neuron, but over long periods this activity should be proportional to the sum of the input weights, resulting in similar synaptic changes.

The CA3 subregion of the hippocampus seems to be involved in several tasks such as storing temporal sequences and associations between parts of memories, while maintaining a homeostasis over the lifetime of individuals. Moreover, it is necessary to operate in the presence of noise, incomplete cues, and limited connectivity. With this study we depicted important properties of the CA3 that can be accomplished using a simple heteroassociative rule combined with competitive synaptic scaling. We believe that both auto- and heteroassociative rules have their roles in memory processes and simulation studies like the one presented here allow us to better understand the effects of different learning rules over the operation of these networks.

## ACKNOWLEDGEMENTS

The authors would like to thank Universidade Federal do ABC (UFABC) for providing the computational infrastructure for the simulations and the members of the Timing and Cognition Lab (http://neuro.ufabc.edu.br/timing/), at UFABC, for their suggestions during project development. We also thank the reviewers of the paper for providing insightful contributions.

### Funding

The authors received no funding for this work.

### Competing Interests

The authors declare there are no competing interests.

## Author Contributions

- Raphael Y. de Camargo conceived and designed the experiments, performed the experiments, analyzed the data, contributed reagents/materials/analysis tools, wrote the paper, prepared figures and/or tables, reviewed drafts of the paper.
- Renan S. Recio conceived and designed the experiments, reviewed drafts of the paper.
- Marcelo B. Reyes conceived and designed the experiments, contributed reagents/materials/analysis tools, reviewed drafts of the paper.

## Data Availability

de Camargo, Raphael (2017): Heteroassociative storage of hippocampal pattern sequences in the CA3 subregion—raw data and simulation scripts. figshare.
https://dx.doi.org/10.6084/m9.figshare.5378347.v2.

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
