# Peer review of "Heteroassociative storage of hippocampal pattern sequences in the CA3 subregion"

_PeerJ, doi:10.7717/peerj.4203_

## Round 0.1 · original submission · Major Revisions

· Academic Editor

Major Revisions

All three reviewers provided extremely thoughtful and thorough reviews, and all three found the study to be interesting, relevant, and largely sound. However, all three identified major areas of the presentation that would need to be addressed in order for the paper to be ready for publication. I will not list all the specific comments below, but I will summarize a few major points.

1. All three reviewers identified major aspects of the methods (particularly modeling details) that are incompletely specified in the paper. These include specific features of the model (e.g. number of OLM and basket cells, voltage threshold etc.), justification for specific choices that appear to vary from the "biological" range, and details of the implementation (e.g. programming language and/or simulator used). One particular point raised by multiple reviewers is the choice not to include NMDA receptors in the model, given the focus on memory and learning rules in the article. I will not re-list all the other points here, but refer the authors to the specific reviewer comments, all of which should be addressed in a revision.

2. As noted by Reviewer 1, in several places you refer to "significant" results obtained with your model. It is unclear to me whether you in fact ran statistical tests to justify these conclusions. If so, please provide full details of statistical tests, including test name, test statistic (e.g. t or F), df, and p-values. If statistical software was used, please give details in Methods section. But if formal statistical analysis was not conducted, please refrain from claiming that manipulations do/don't "significantly" change the results. Please also clarify whether the results presented are taken from a single simulation run or represent averaged behavior over multiple runs – and, in either case, please clarify how much variability was present between runs. Currently, I can only find this information for Figure 6 (and even there, variability is inferred graphically).

3. Reviewer 3 took issue with the use of "STDP" to describe the LTP and LTD-like processes in your model. Please carefully consider how you would respond to these comments. One option would be dropping the claim that the plasticity mechanisms in your model capture/simulate STDP (although you could note that your learning rules were "inspired by STDP"); alternatively, please add additional justification to describe why the plasticity mechanisms in your model adequately capture STDP. (I would suggest the former, in that it appears the important point in your article is contrasting presence/absence of LTP/LTD, rather than making a strong claim that this does/doesn't adequately capture all the features of STDP.)

4. The reviewers also carefully noted a number of language/presentation issues that should be corrected, and made specific suggestions for improving the clarity and content of the abstract and key hypotheses, as well as clarifying some existing ambiguity in key terms, particularly in the introduction. (Most importantly, perhaps, is clarifying the difference between autoassociative and heterassociative memory, and between sequence completion and pattern completion.) These specific notes appear both in the reviewer comments and in the annotated manuscripts provided by Reviewers 1 and 3.

While the points raised during review are numerous and extensive, most involve re-writing of the article; all three reviewers found the underlying work itself to be sound, and the model well-constructed and interesting (and I concur). I encourage you to consider revising and resubmitting the article. If you decide to prepare a revised version, please note how you have responded to each point made by the reviewers. I will then send the article back to these reviewers for re-review.

·

Basic reporting

In this study de Camargo et al. investigate whether a simple learning rule is sufficient to explain the features that make the hippocampus a suitable candidate for memory formation, storage and retrieval. The authors state that they have used a biologically plausible network model of the CA3 hippocampal subregion consists of 10,000 pyramidal cells accompanied with fast and slow inhibition in a feed-back manner. The authors have found that a network with synaptic weights determined by a heteroassociative learning rule can store and retrieve patterns and also renew the set of already stored memories with new ones. In addition, combining the heteroassociative learning rule with a homeostatic mechanisms, i.e., synaptic scaling, have permitted the storage of new memories while forgetting older ones. The motivation and the findings of this work are interesting. The network they used consists of 10,000 neurons which means that is biologically plausible in terms of neuronal numbers. This work is an accurate exploration mainly three parameters: the sparsity, the connectivity extent and the use or not of the modified STDP rule. Overall, this work seems to me that it is very interesting and well explained. In addition, it is well written and its structure meets the rquirements of this journal. However, I have some concerns about the modelling approach and the findings presentation.

Experimental design

The Methodological part is in general very informative, however it is hard to replicate this network as some critical parameters are missing (see below for details).

Validity of the findings

The findings are clear presented and the figures are in the majority of the cases accurate and correct. However, if there is a weakness, it is in the lack of any statistical analysis of the results. I suggest that you provide all the statistical tests you have used and also you should report their results (p-values, statistical values etc.). Finally, the conclusions are very informative and they give insights of the significance of this work.

Comments for the author

Generally, I suggest that you define in the Introduction more clear the difference between hetero- and auto-association memories. If anyone is not familiar with this topic it would find it difficult to follow.
In addition, in my point of view it is of high importance to provide the code that is used on ModelDB or any relevant website.
Also, Figures 4a and 4b never mentioned in the text either explicitly or as Figure 4 in general.
Finally, it is better to use a larger line-spacing while you are in a review process as it would be easier to read.

Following are specific comments on methodology part (not in order of significance or appearance in the text):

1. It is widely known that CA3 receives two major inputs from DG via mossy fibers and from EC LII and their role have been studied (Treves, Rolls, 1992). The authors should justify why the do not simulate explicitly the inputs.

2. Related to my previous concern, the authors state that their model is "a biologically plausible model". However, they do use a very simple form of inhibition in the network. I am wondering if by using fedd-forward inhibition as well would eventually change their findings.

3. Using integrate and fire neurons, it is necessary to report the voltage threshold as well as the reset voltage after a spike.

4. NMDA receptors are very important in pyramidal cells and they play a crucial role in memory procedures. There is no justification of the reason they are not simulated in the network.

5. I would like to know how many OLM cells are in the netowork and how many the Basket cells. In addition, in Figure 1 the name Basket appears but there is not in the main text.

6. Justification on the parameters choice is missing. In addition, the reason that the authors have chosen such a large refractory period is not explained.

7. Line 147-148: It is not explained how the difference is subtracted from the rermaining synapses. Randomly? Uniformly?

8. Critical simulation parameters and information should be provided, such as the total simulation time, the programming language and/or the simulator that use.

Following are specific comments on the results validity (not in order of significance or appearance in the text):

1. Although Figure 6 is one of the key finding of this work, I believe that the way the findings are presented in this graph is somehow confusing. I highly suggest this figure to be re-made.

2. In Line 251, they say that "retrievable patterns increased significantly", however there is no reference to any statistical test in the whole paper. I highly encourage the authors to report the statistics af any comparsion they have done, as well as to report which test is used in every case.

3. Typo in sparsity legend in Figure 2 (right side, on the top)

·

Basic reporting

de Camargo et al. studied the performance of a heteroassociative coding scheme, combined with synaptic scaling, in pattern sequences storage in a biologically plausible model of the hippocampal CA3. They looked into pattern learning of the network, the ability to retrieve stored pattern sequences upon presentation of a partial cue, as well as the renewal of stored sequences as new memories are learned. The objectives of the research article are sound, and the report of the findings is also clear.

While the authors discuss a lot about the pros and cons of heteroassociative and auto-associative networks, it would be good if more background on their definition and their network architectures could be provided. It is unclear to me how “to combine the 30 properties of auto- and heteroassociative networks” (line 29-30), and why “feedback connections from CA3 to Dentate Gyrus or between different CA3 areas would store sequences of memories in heteroassociative connections, and recurrent connections within single CA3 areas would work as auto associative connections” (line 32-34). On the other hand, the authors want to show that a heteroassociative coding scheme would suffice to account for the features that make the hippocampus a reliable network for memory.

In line 36, it is not clear what the authors want to say with "representing sequence of neuronal ensembles which are sequentially activated bounded by theta cycles”.

In line 71, it is better to use the term "oriens lacunosum-moleculare” once and then O-LM later as abbreviation.

In line 190, STDP-induced LTD is abbreviated as SS-LTD, but in the context SS-LTD seems to refer to synaptic scaling combined with STDP-induced LTD. Please clarify.

Experimental design

The design of the model fits well to the aim of the study. Most part of the network model is clearly described. However, the authors shall verify the model scheme and the use of parameters.

Why are NMDA conductances not considered in the model? Earlier work on CA3 suggested that CA3 exhibits attractor dynamics due to the abundance of recurrent excitation, and NMDA conductances are needed to support attractor dynamics.

In line 67, "The network has two kinds of feedback inhibition, mediated by fast and slow GABA channels (Pearce, 1993), modeled as direct connections between every pair of pyramidal neurons.” There is basket cell in Figure 1 but the inhibition is implemented as direct interaction between pyramidal cells. It would be helpful to the readers if you add a model scheme of how exactly the model is implemented, and include all kinds of inputs the CA3 circuit receives.

In line 75, does “neuron” refer to pyramidal neuron?
In line 77, a membrane time constant of 2ms is a bit small. Can you justify it?
In the equation below line 79, is the negative sign missing in the exponent?
In line 83, the step size is set to 0.1ms, which is a bit big compared with a membrane time constant of 2ms. Have you verified that the simulation results converge?
In line 106, is the external noise between neurons independent?
In line 109, a rate of 0.8 Hz during pattern retrieval in the simulations seems to be slow, compared with the firing activity in Figure 4.
In line 115, as the computational demand is not that high for point neurons, it will be interesting to check the memory capacity and performance as the network size is varied.
In line 144, could you add one or two sentences to describe what heterosynaptic long-term depression is?
In line 145, is the additive synaptic scaling rule the same as the usual normalization? Why the rule is applied after 100 sequences, not after every sequence?
In the formula below line 172, what if all neurons o_i(t) are active? Shall it be normalized by sqrt(n_p*n_o) instead?

Validity of the findings

The report of the findings is clear in general. There are a few minor issues that the authors should address.

In line 185, "storing a pattern sequence increases the total synaptic weight of each neuron, which is compensated by the synaptic scaling rule”. Is it true only if you have no or weak LTD?
In line 258, the connectivity extent of 0.4 to 1 looks a bit high. What is the biological range?
In Figure 4 right column, there are 10 peaks in 4(e) but 7 patterns are used. Could you explain?
What is the function and advantage of theta rhythm in the model, does it work without theta rhythm?
In line 211, "the benefits of using less neurons per pattern” are stated. How about the disadvantages? The benefits come with expenses.

Comments for the author

I think the article by de Camargo et al. could be accepted after addressing the comments above.

·

Basic reporting

The language of the article as written is mostly fine and understandable, but there were some sticking points for me as a reader that confused me until I was able to infer the intended meaning much later in the paper. (My annotated copy details my struggles as I read through from the beginning.) A major ambiguity for me was “connectivity extent”, as I thought at first “extent” was used as “level” or “value”, in which case I would drop “extent” altogether and just refer to “connectivity”. However, it became apparent that it meant the initial connectivity of the network, specifically the initial fixed fan-in connectivity fraction per neuron. Since “final connectivity” is referred to later in the paper to indicate connectivity after learning, I would strongly suggest replacing “connectivity extent” with “initial connectivity” and expand the methods definition to make the distinction more explicit.

Another major ambiguity was “sequence completion” versus “pattern completion”: this paper deals only with sequence completion yet uses the term interchangeably with pattern completion. I take “pattern completion” to indicate autoassociative dynamics that retrieve a full pattern based on a degraded cue, so for much of the paper I was waiting for the authors to demonstrate autoassociative processing in their model. I marked instances in my annotated copy, but Lines 116, 294, 296, 383, 388 are examples. You’ll note I figured out this was a purely heteroassociative model at Figure 4 where the full cue pattern is not itself retrieved (the overlap curve is flat for the cue patterns in Figure 4e).

The abstract and introduction refer to the “coding scheme” being studied (Lines 11 and 51), but to me this is a very ambiguous phrase and I didn’t understand what was being referred to. It seems that the authors meant to refer to heteroassociative processing as a whole as a “coding scheme”. Under that interpretation, the main hypothesis of the paper translates to whether heteroassociative learning (with homeostatic synaptic scaling) suffices to account for the role of the hippocampus in memory, with the implication that the alternative “coding scheme” of autoassociative dynamics may be unnecessary. If that is the case, the results of the simulation study fall far short of supporting the stated hypothesis. This is not an issue with the scope of the simulation study, more that the hypothesis interpreted this way is, I believe, overly and unnecessarily broad. I would like to see “coding scheme” replaced with a more specific and clear phrase and the hypothesis narrowed down to the more specific questions addressed by the study.

Another ambiguous term was “renewal” used in the context of “renewing the set of stored sequences”. I did not understand what was being “renewed” in this usage, but came to sense the authors meant the abstract set of currently available (or “stored”) sequences was “refreshed” or “updated” as new sequences replaced old sequences. I originally thought “renewal” meant that old sequences were somehow reinforced, which I believe is the opposite of the intended meaning. Given its key place in the abstract and introduction (Lines 49 and 52), I would suggest replacing “renewal”/“renew” with clearer terminology (whether “updating”/“update” or similar would suffice I’m not sure).

A number of sentences do not follow the scientific logic of the results and seem to editorialize unnecessarily. For instance, in arguing for “heteroassociative” processing, the authors seem to go to lengths to argue against autoassociative dynamics, for which there is substantial evidence and which do not need to be repudiated to discuss heteroassociative effects in sequence learning. This is further reinforced by the broad and dichotomous hypothesis in the abstract as I mentioned above. There are autoassociative effects of hippocampal memory that cannot be explained by the sort of heteroassociative network studied here (cf. the flat cue pattern retrieval in Figure 4e).

“Biologically plausible” (in abstract and Line 59). I really do not like this phrase, it is overused by modelers (including me in the past!) to the point it doesn’t mean anything; I much prefer to see specific language supporting why the details of the model constitute an appropriate approach to the scientific question under study rather than this blanket assertion of “plausibility”.

I object to the consistent description of LTD throughout the paper as “STDP-induced” or “STDP-based” LTD. There is no STDP in the network training. The authors describe a time-windowed learning rule perhaps inspired by STDP as indicated by the citations to the STDP literature, but it is not an actual implementation of STDP, and it is wrong to describe the LTD rule that way. The optional LTD in the learning rule is simply a time-windowed heterosynaptic LTD.

The breadth of literature references is generally fine, if overly dependent on John Lisman’s work. I would suggest that W. B. Levy’s studies of hippocampal sequence learning would provide additional valuable and instructive background, particularly his 1996 paper (Hippocampus 6, 579-90).

The structure of the paper is fine. However, the abstract does not do enough work. In addition to my comments above on the abstract, it seems to end at the point where the results of this study should be summarized and put into context for the readers. As it is, the abstract provides (too much) background, an overly broad hypothesis, and a simplistic description of the “biologically plausible” model. The entirety of the results are captured by the phrase “these hypotheses hold”, but there is no elaboration of actual results, how they support the hypotheses, any problems, or what the take-away conclusion of the study should be. This abrupt end to the abstract is a disservice to potential readers of the paper.

Given the above caveats with the scope of the hypothesis, the study is self-contained, relevant to the questions posed, and constitutes an appropriate unit of publication.

Minor points of language:

- Line 17, the definition of “autoassociative memory” is wrong, referring to “association between different memories” when it should be “association between the features of a given memory” (or similar)
- Lines 42 and 425, I believe “temporally” was supposed to be “temporarily”
- Line 36, the phrase “Different from” should be “In contrast to” in idiomatic English
- Line 28, “representing” should be “represented by” unless my understanding is wrong
- Lines 230 and 238, “bifurcation” is the wrong word, maybe “transition” or “maximum”
- throughout, “less” is used when “fewer” is correct (with discrete noun plurals), I’ve marked most instances in the annotated copy. Ibid. with “smaller” and “lower” for numerical values
- Line 42, “third feature”. It is unclear how the authors are counting the features of the hippocampus being introduced; I didn’t realize there was a “first” or “second”
- Line 48, “acceptable”. Saying an effect is “acceptable” for CA3 is presuming fundamental knowledge of the actual purpose and role of CA3; this is tautological and teleological. One could say it is “consistent with” a “hypothesized role” of CA3.
- Lines 205 and 207, the authors never define “crosstalk”. Is it a form of memory interference? Does it mean anything beyond “memory pattern interference”?
- Line 253, “neurons per pattern” doesn’t make sense to me, did the authors mean “connections per neuron”? I don’t think the pattern size was changing there.
- Lines 413-415 and 426-428, I’ve highlighted these sentences in the discussion as being vague and unclear. They should be elaborated more clearly or dropped.
- Line 345, missing noun after “autoassociative”
- Line 422, missing noun after “heteroassociative”
- Line 435, missing “are”

Experimental design

This a primary research manuscript pertaining to Biological Sciences, which is appropriate to the Aims & Scope of the journal.

I do have concerns with the statement of the research question, as I discussed in the Basic Reporting section. It is overly broad and dichotomous, forcing an unnecessary counterpoint to autoassociative learning when the history of research in this area suggests that these dynamics are not mutually exclusive. I would like to see a narrowing of the scope of the research question in a way that makes it more specific and attuned to the subtleties of homeostasis, synaptic modification, and temporal interactions in sequence learning that are actually probed in the simulation studies.

After reviewing the methods descriptions and a shallow review of the source code, the technical standards for the simulation study are in-line with the typical research output in this field of study (spike-based networks of sequence learning with various synaptic modification rules).

For a simulation study like this, the only route to reproducibility is source code release and it appears that the source code is available (or will become available) on figshare with a DOI.

However, the authors’ methods descriptions have a number of minor issues. My annotated copy is complementary to the points below.

Network model:

- The Figure 1 caption refers to the inhibitory “neuronal populations” but the description and figure graphic suggest that a single global interneuron (basket cell) is used to control activity. Line 71 mentions “O-LM” cells for inducing theta oscillations, but not how many are in the model, or why more than one are needed since they would presumably have the same activity. All of this should be clarified.

- Line 68 describes the feedback inhibition as “direct connections between every pair of pyramidal neurons”, but that would constitute feedback excitation. Something is wrong here, unless it was implied that the “direct connection” was through the global interneuron, which is something that should be made explicit.

- Figure 1 shows the O-LM interneuron driven by the pyramidal neuron output, but it is described in the text as a spike generator. The figure should be fixed to remove those pyramidal/O-LM connections if they don’t actually exist.

- Line 72. It is unclear what “intrinsic” firing rate means or why the theta frequency of 5 Hz was chosen. That would be considered atypically slow theta in rodent hippocampus

- Line 72, “It” (referring to one or multiple O-LM cells) is said to “modulat[e] the generation of the theta rhythm”, but it IS the theta generator, so it is unclear what modulation is being described here

- Line 79, “neural adaptation” is too broad and should be specified as, I believe, “spike frequency adaptation”

- Irep equation should have exp(tspk – t) instead of exp(t – tspk), otherwise the hyperpolarizing current would be exponentially increasing with time

Synaptic model:

- Ik equation should have a halfwave-rectified \Delta{t_s} otherwise it would be negative after the spike but before the delay has finished

Pattern sequences:

- Line 111, the authors should specify whether pattern sampling is with or without replacement. The patterns should be orthogonal, which would be sampling without replacement. If not orthogonal, justification will be needed

- Line 113, the definition of “sparsity” is actually a definition of “density”. Sparsity should be defined as the fraction of inactive cells in the pattern (the authors define it as the fraction of active cells). Instead of changing the numbers and figures, it would be easiest to change “sparsity” to “input density” (or similar) throughout the paper to convey the meaning correctly.

- Line 124, typo “s1-s2-s2-…”

- Lines 115-118, the authors justify a higher input density than de Almeida’s 2007 estimates for their 10,000-neuron simulations, but it would be nice to see an analysis of larger simulations (e.g. 30K neurons) that may allow sequence learning with more “realistic” input densities to see if network size was the limiting factor

Learning rule:

- Line 132, 15-ms presentations for off-line training seems to presume that the sequence has already been learned, in order to compress events into gamma cycles. I suppose the authors are considering an ordered set of place cells that fire in sequence within a theta cycle due to phase precession. However, this should be clarified in text and made explicit to readers with and without background knowledge of theta sequences

- Weight update equation is unclear. How is the initial ± supposed to be interpreted? Is there a forward pass for LTP followed by a backward pass for LTD?

- Also, weight update is clearly not based on spike-timing differences. As I mentioned above: it is not an STDP rule, it is only a sequence-adjacency rule. This approach is valid for studying the parameters of sequence learning, but it is not valid to refer to this as “STDP-induced” LTP/D throughout the paper. STDP can have complex consequences and effects that are not accounted for with this rule.

- Line 142, I think this is the beginning of my “connectivity extent” problem described above. I did not understand why the authors related the initial connectivity to that of a “highly connected developing nervous system”. New episodic memory sequences are not learned from an initial state corresponding to a developing nervous system. The authors should clarify that they train these networks from a highly connected state with many sequences until stabilizing at a lower connectivity, effectively simulating a “lifetime” of episodic sequences until maturity. More clarity about this training regime would help.

- Lines 153-154, “by anatomical or physiological reasons” is confusing. I’m not sure what it means since the model only contains synaptic connections

Validity of the findings

The findings of this simulation study have high face validity: I believe the results are relevant to the research question(s) that the authors aimed to study. Further, the paper as presented demonstrates substantial content validity: Many of the parameters most relevant to the narrow scope of “pure” heteroassociative learning in a network model are studied in an interesting and inter-related way that helps to bridge a knowledge gap in the combination of homeostatic and experience-dependent forms of synaptic modification in a cognitive neural system. The authors’ lack of consideration of autoassociative processing in this system reduced the potential content validity, since there may be critical interactions (e.g. with other hippocampal inputs or subregions) that contribute to sequence learning as such and were not addressed by this work. However, it is good for the field to have studies like this that explore particular dynamics of interest.

The data, consisting mostly of static network analysis and network simulation outputs, appear to be robust and statistically sound within the standards of the field.

As with the abstract, which ended abruptly, the paper does not conclude in a way that summarizes the results or relates those results back to the specific research questions. The authors should consider trimming some of the vague discussion and adding a strong conclusion that ties specific results back to specific knowledge gaps revealed by the research question. Given my concerns about language and structure above, to make this possible the research question will need to be reframed and better specified throughout at least the abstract and introduction in any subsequent revisions.

---

## Round 0.2 · Minor Revisions

· Academic Editor

Minor Revisions

I have now received comments from all three reviewers on the revised version of your article, and all three reviewers agree that the revised version is much improved and now almost suitable for publication -- although they have identified quite a few remaining issues with language and readability. These are mostly line edits related to language usage, but a few are requests for clarification. Most should be straightforward to implement. Please note that two reviewers have submitted annotated pdfs with suggested line edits. In most cases, I agree that the reviewers' suggestions would substantially improve readability and clarity.

As noted above, if you can address these comments and resubmit, the manuscript will mostly likely be accepted without need for further re-review.

·

Basic reporting

The language is clear and not ambiguous. After revisions the authors have taken into account my comments and more specific, they give a clear definition of auto- and heteroassociative networks. In addition, the authors have given a link of a webpage where anyone can download the source code.

The only disagreement with the authors is that even if the learning process is performed online, in my opinion, NMDA current should be included in the model as it is widely know that plays an important role in dendritic integration in pyramidal cells in CA3 and CA1. However, for the specific questions that the authors have tried to answer it is valid to ignore this parameter in order to reduce the parameter space. It would be interesting though to see if any change would arise by incorporating the NMDA component to this model.

Taken all into account, I think that this project is almost ready for publication.

However, I have some minor comments about the construction of the figures. In my opinion it is better to produce graphs without top and left axes (Figure 6 axes are nice) for consistency and clarity.

Moreover, in Line 197, where authors refer to MATLAB I think that you should include also the version and also the computer specifications (CPUs, RAM) where the code has run. Typically, MATLAB (R2016a, The Mathworks, Natick, MA) form is used.

Additionally, some minor suggestions are given below:

Line 33: missing noun in "autoassociative"
Line 78: "send connections ..." with "is connected with every pyramidal neuron via GABA synapses..."
Line 84: "is dependent" with "depends"
Line 125: "to mean that" with "to refer to"
Line 219: "would be performing" with "would be"

Experimental design

no comment

Validity of the findings

no comment

Comments for the author

no comment

·

Basic reporting

The authors have updated the manuscript according to reviewers' feedback on language use. The manuscript is in general well-written. However, there are still some issues with the use of words.

In line 26, replace “the cell preferred place” by “the cell’s preferred location”.

In line 51-52, for “since an increase in the number of stored memories leads to a catastrophic interference effect”, do you mean an increase beyond the capacity limit?

In line 155, “We decrease all synapses from each neuron by a fixed amount”. I guess the authors mean to reduce all synaptic weights.

In line 279-280, “For the initial random weights, we used the values of 2.5, 280 3.0, 6.0 and 4.0 for the initial connectivities 1.0, 0.8, 0.6 and 0.4, respectively” is confusing.

In line 301, “larger then” should be “larger than”.

In line 329, “latter” should be “later”.

Experimental design

In the METHODS, the authors reported the learning rule first and then connectivity. It seems more natural to describe the network connectivity before the learning rule. The readers want to know how the network is like before what changes to make.

In line 92, shall it be for all t > ts + tdelay?

In line 194, what are the combinations of parameter values?

Validity of the findings

In line 266-268, "If we consider the estimated connectivity of 20%, which full connectivity we would have to use an initial weight of 2.0, resulting in about 1400 retrievable patterns.” Did the authors mean “with full connectivity”? Also in line 268-271, the description is not totally clear. Are the readers supposed to look for the initial weight corresponding to 20% connectivity from Figure 3A given the initial connectivity, and then figure out the number of retrievable patterns from Figure 3B with the corresponding weight and the given initial connectivity? If so, it would be helpful to indicate the 20% connectivity and mark the values used for comparison in the figures.

In line 411 and 413, what are errors refer to?

Comments for the author

There are some very long sentences in the article. For example, in line 15-19, “Based on these models, Rolls et al. (1997) proposed that the CA3 subregion could work as an autoassociative memory, where neurons from the same pattern have recurrent excitatory connections between them, which enable associations between features of a memory, retrieval of stored memories from presentation of partial cues, and noise tolerance during memory retrieval.” Could the authors consider breaking them down to improve readability?

It seems that the authors dropped “extents” following the feedback from Reviewer 3, but there are still a few occurrences (e.g. in line 347).

·

Basic reporting

The authors substantially improved the clarity of the text and the results. The revision and letter demonstrated a comprehensive treatment of points from the initial review. The arguments about auto- vs. hetero-association have been smoothed out and provide better context, including additional references, and the central question of the study was more prominent and more clearly defined. The abstract and discussion now have proper conclusions that will certainly help readers. I believe the manuscript now meets the journal’s criteria for publication.

There were a few places where “extent” was still used instead of the new “initial connectivity”. There were also some minor typographical issues that I have marked in my review copy that should be fixed prior to publication.

Experimental design

The model design, especially regarding the implementation of interneurons and inhibition, has been given appropriate detail and clarified in the network figure. The expanded details about the number of simulations, means, and standard deviations in the figure legends has increased the transparency about the study design and findings.

Validity of the findings

As in my initial review, the modeling study has both face and content validity for the central question of the paper. The additional language about the relationship with auto-associative processing established a narrower focus that helps to allay my prior concerns about overly broad claims.

---

## Round 0.3 · accepted · Accept

· Academic Editor

Accept

I have had the opportunity to carefully review your revised manuscript, and I am satisfied that you have responded to all the reviewer comments. I am pleased to accept your article for publication in PeerJ. I believe the article describes a very interesting series of simulation studies that will make a relevant contribution to the literature. I hope you will consider PeerJ as a potential outlet for future submissions.